# Cystic Fibrosis-Related Diabetes (CFRD): Overview of Associated Genetic Factors

**DOI:** 10.3390/diagnostics11030572

**Published:** 2021-03-22

**Authors:** Fernanda Iafusco, Giovanna Maione, Francesco Maria Rosanio, Enza Mozzillo, Adriana Franzese, Nadia Tinto

**Affiliations:** 1Department of Molecular Medicine and Medical Biotechnology, University of Naples “Federico II”, 80131 Naples, Italy; iafusco@ceinge.unina.it (F.I.); maione@ceinge.unina.it (G.M.); 2CEINGE Advanced Biotechnology, 80131 Naples, Italy; 3Regional Center of Pediatric Diabetology, Department of Translational Medical Sciences, Section of Pediatrics, University of Naples “Federico II”, 80131 Naples, Italy; francescomr@hotmail.it (F.M.R.); enza.mozzillo@unina.it (E.M.); adriana.franzese@unina.it (A.F.)

**Keywords:** cystic fibrosis, hypoglycemia, hyperglycemia, diabetes, cystic fibrosis-related diabetes (CFRD), glucose metabolism alterations

## Abstract

Cystic fibrosis (CF) is the most common autosomal recessive disease in the Caucasian population and is caused by mutations in the CF transmembrane conductance regulator (*CFTR*) gene that encodes for a chloride/bicarbonate channel expressed on the membrane of epithelial cells of the airways and of the intestine, as well as in cells with exocrine and endocrine functions. A common nonpulmonary complication of CF is cystic fibrosis-related diabetes (CFRD), a distinct form of diabetes due to insulin insufficiency or malfunction secondary to destruction/derangement of pancreatic betacells, as well as to other factors that affect their function. The prevalence of CFRD increases with age, and 40–50% of CF adults develop the disease. Several proposed hypotheses on how CFRD develops have emerged, including exocrine-driven fibrosis and destruction of the entire pancreas, as well as contrasting theories on the direct or indirect impact of *CFTR* mutation on islet function. Among contributors to the development of CFRD, in addition to *CFTR* genotype, there are other genetic factors related and not related to type 2 diabetes. This review presents an overview of the current understanding on genetic factors associated with glucose metabolism abnormalities in CF.

## 1. Introduction

Cystic fibrosis (CF) is the most common autosomal recessive disease among Caucasians, affecting about 1/2500 newborns [1,2]. It is caused by mutations in the gene encoding the cystic fibrosis transmembrane conductance regulator (CFTR) protein, a chloride channel expressed in several tissues, such as respiratory, gastrointestinal, biliary, hepatic, pancreatic, and reproductive tissues. CFTR is responsible for the outward movement of Cl^−^ and the linear movement of water across cell membranes. This channel regulates other apical membrane-associated channels including the epithelial sodium channel, the outwardly rectifying chloride channel, and the retinal outer medullary potassium channel [3,4,5]. Therefore, a CFTR dysfunction determines an alteration of the ionic balance of physiologic secretions, with subsequent formation of thick and viscous secretions causing inflammation, obstruction, and destruction of ducts in the lungs, pancreas, liver, and reproductive organs [1,6]. CFTR also regulates bicarbonate secretion, especially in the pancreas, intestine, and lung [7], and it is believed that, in CF, the reduction in bicarbonate is the main cause of alterations in these organs [8,9].

Mutations in the *CFTR* gene can cause different alterations of the protein: its synthesis and stability, its transport to the apical membrane, its gating, and its conductance of ions through the channel [10]. Up to now, over 2000 variants of this gene have been identified worldwide [11], over 300 of which are known to cause CF. These variants are categorized into six classes according to their specific effects on CFTR function [12]. Classes of mutations I, II, and III tend to have more severe consequences, whereas classes IV, V, and VI cause minor changes in the protein function leading to a milder phenotype [13,14]. The F508del mutation (class II) is the most common pathogenic variant and is characterized by a deletion of phenylalanine at the 508 position that provokes a conformational alteration of the protein, leading to decreased expression of CFTR at the cell surface [15]. However, it is not possible to stick to this classification alone, because many *CFTR* mutations cause consequences for the protein that fall into two or more classes, hindering a clear separation between the various categories [16]. The phenotype and evolution of the disease are also influenced by other genes, called modifiers of the *CFTR* gene [17]. These genes, inherited independently from the *CFTR* gene, can interact with the latter, increasing or reducing its function with variable consequences on clinical expression [18].

Lung disease is the leading cause of morbidity and mortality in CF, and it is associated with the development of comorbidities and chronic infections [17,19]. Among the extrapulmonary comorbidities, abnormalities of glucose metabolism play a primary role. Cystic fibrosis-related diabetes (CFRD) is the most prevalent extrapulmonary comorbidity in CF [2] and is associated with more frequent pulmonary exacerbations, fast deterioration of lung function, and higher mortality [20], whereas hypoglycemia rarely occurs [21]. CFRD prevalence increases exponentially with age, affecting 2% of children, 19% of adolescents, and 40–50% of adults with CF [22,23], with a higher prevalence in females [22,24]. As observed, although CFRD is uncommon before puberty, abnormalities in glucose tolerance may be present in CF from infancy [25], with a higher risk of early onset of diabetes in these children [25,26].

CFRD is recognized as type 3c (pancreatogenic) diabetes, and, despite displaying characteristics common to both type 1 diabetes (T1D) and type 2 diabetes (T2D), it represents a distinct clinical condition with pathophysiological and clinical differences which require a specific approach for diagnosis and management [27] (Table 1). CFRD is associated with some of the same long-term complications of T1D and T2D; however, the more important complications are a more rapid decline in lung function, poorer nutritional status, and reduced survival [4], which have been shown to improve with treatment of CFRD. Therefore, diagnosis and appropriate treatment of CFRD are crucial components of the medical care of a CF-affected patient.

### 1.1. Clinical Presentation and Management

Improvements in CF life expectancy and in diabetes screening practices have led to an increase in the prevalence of CFRD. CFRD is associated with worsening of the underlying disease that begins to develop years before diabetes is diagnosed. The literature shows that, in CF patients with prediabetic glucose alterations who later developed CFRD, there is a decline in clinical parameters (respiratory function and nutritional status) 6 years prior to diagnosis of diabetes [29].

Transient hyperglycemia may initially occur only during periods of acute lung disease or during treatment with corticosteroids. With the progression of the disease, asymptomatic postprandial hyperglycemia appears, detectable only by an oral glucose tolerance test (OGTT), presenting the clinical picture of overt diabetes.

Severe genotypes, pancreatic insufficiency, and the female gender are independent risk factors for early glucose derangements in CF [30,31]. CFRD has also been reported in young patients, including infants [32]. T1D, which is not associated with CF, has been described as an additional disease in some CF children; therefore, it should be suspected in CF children with diabetes diagnosed at a young age [33].

CFRD is often asymptomatic and begins insidiously, remaining undetected for a long time. Onset with diabetic ketoacidosis is exceedingly rare. More frequently, this form of diabetes can be diagnosed in the phases of acute lung infection or corticosteroid therapy when there is a worsening of insulin resistance.

Some of the symptoms of diabetes, such as polyuria and polydipsia, can be partly confused with those of CF. CF patients often drink heavily due to a dry mouth feeling. Other symptoms of CFRD may be feeling tired, weight loss or inability to maintain ideal weight despite a high energy intake, poor growth rate, and pubertal delay. Furthermore, worsening of respiratory function and an increase in the frequency and severity of respiratory exacerbations may be presenting signs.

### 1.2. Diagnosis and Screening

According to the American Diabetes Association (ADA) [34] and the International Society for Pediatric and Adolescent Diabetes (ISPAD) guidelines [35], the following diagnostic criteria of CFRD have been established, which are present during periods of stable health conditions: 2 h blood glucose at T120′ of OGTT >200 mg/dL (>11.1 mmol/L), fasting blood glucose (FPG) >126 mg/dL (>7 mmol/L) on two or more occasions, glycated hemoglobin (HbA1c) ≥48 mmol/mol (6.5%) (48 mmol/mol), or random blood glucose >200 mg/dL (>11.1 mmol/L) on two or more occasions with symptoms. HbA1c below the aforementioned value does not exclude the diagnosis of CFRD. During flare-up phases of the disease, where intravenous antibiotic therapy and/or systemic corticosteroid therapy are required, the diagnosis of CFRD can be made if, for 48 h, FPG >126 mg/dL (>7 mmol/L) or postprandial blood glucose >200 mg/dL (>11.1 mmol/L) is present.

Due to the insidious methods of onset of CFRD, the annual screening examination through OGTT is crucial in identifying affected or high-risk subjects. The OGTT is currently the screening test indicated by the latest guidelines and must be performed once a year in patients aged >10 years, as shown in Table 2 and as reported in the ISPAD guidelines [28], which shows how to classify glucose metabolism disorders in cystic fibrosis.

A prospective longitudinal study conducted on children with CF between the ages of 6 and 9 years demonstrated that impaired glucose tolerance (IGT) and indeterminate glycemia (INDET) conditions predict risk of developing CFRD during adolescence. For this reason, in some centers, screening is performed from the age of 6 years [25,36].

### 1.3. Pathophysiology

Similarly to T1D, in CFRD, the primary defect is insulin insufficiency [6]; however, CFRD is not an autoimmune disease because the presence of both diabetes-associated autoantibodies and human leukocyte antigen (HLA) susceptibility alleles is not different to the general population [14,37]. As with T2D, CFRD shows an increase in prevalence with age, a progressive defect in beta-cell function, and a modest and fluctuant insulin resistance. In addition, several T2D susceptibility genes have also been evidenced to be in association with CFRD [6,23]. The pathogenesis of CFRD is complex and multifactorial [1] (Figure 1).

Mechanisms that determine CFRD development are not completely defined, but partial insulin deficiency, secondary to the destruction or impairment of pancreatic islets, appears to be the main defect [38]. In CF, pancreatic involvement is characterized by ductal obstruction, due to the hyperviscosity of exocrine secretions, fatty infiltration, and progressive fibrosis, along with destruction of the exocrine pancreas and loss of pancreatic islets [39]. The remaining islets result altered and disorganized by fibrotic and fatty tissue bands [40], contributing to the abnormal endocrine pancreatic function and to insulin insufficiency in CF patients. Although pancreatic fibrosis is considered the major cause of beta-cell destruction in CF, not all CF patients with pancreatic fibrosis develop CFRD, and no correlation has been demonstrated between the degree of pancreatic fibrosis and the incidence or the severity of diabetes. Autopsy studies have demonstrated that a percentage of damaged islets do not correlate with development of diabetes [41,42]. In another study, Bogdani et al. reported a reduction in beta-cell mass in pancreatic tissues of very young children with CF in the absence of diabetes, independently from the degree of pancreatic fibrosis, suggesting that a very early beta-cell deficiency, before the development of lipofibrosis, may contribute to glucose intolerance in the CF pediatric population [43].

It is evident that exocrine fibrotic damage alone cannot completely explain the development of CFRD [43,44], and several additional factors may contribute to either beta-cell mass reduction or beta-cell dysfunction: (a) the CFTR defect; (b) genetic factors related and not related to T2D; (c) inflammation; (d) changes in the incretin axis [38].

CFTR seems to be expressed in beta-cells where it would be involved in pancreatic development and would play a role in insulin exocytosis through interaction with the Anoctamin-1 protein (ANO1), a voltage-sensitive calcium-activated chloride channel [45,46,47]. In the literature, it was demonstrated that the maintenance of high intracellular [Cl^−^] in beta-cells is an important determinant in glucose-induced depolarization [48], and that abnormal beta-cell electrophysiological properties due to inhibition of CFTR Cl^−^ channel function would lead to a more hyperpolarized resting membrane potential and decreased action potential, resulting in a slower response of beta-cells to glucose. These results allow us to hypothesize that, in CFRD, a defect in CFTR function could lead to abnormal beta-cell electrophysiological properties underlying insulin secretion [49,50].

As proof of a role of CFTR in human insulin secretion, some authors demonstrated that drugs correcting CFTR dysfunction improve insulin secretion and glucose intolerance after 1–4 months of treatment [51,52]. In addition to beta-cells, CFTR is expressed in alpha-cells [47], where it has a role in glucagon suppression. Huang et al. proposed a model for regulating glucagon release intrinsic to alpha-cells involving both CFTR and K_ATP_ channels. In this model, the authors hypothesized that, in alpha-cells, when CFTR is activated by glucose, probably through glucose-induced ATP changes, by glucose-dependent kinases, or by cyclic adenosine monophosphate, it potentiates K_ATP_ channels, resulting in a strong hyperpolarization that diminishes the Ca^2+^ response required for glucagon release [53]. When CFTR is defective, the potentiation effect of CFTR on K_ATP_ is removed, leading to a membrane depolarization due to K_ATP_ closure, opening of voltage-dependent Ca^2+^ channels, and excessive glucagon secretion [53]. These data support the concept that, in addition to defects in insulin secretion within the beta-cells, dysregulated glucagon secretion, due to CFTR dysfunction in alpha-cells, may contribute to the glucose intolerance in CF and to the development of CFRD [47].

Inflammation likely plays a role in the etiology of CFRD. Recent studies demonstrated the increased production/secretion of multiple cytokines and chemokines such as interleukin 1 (IL1)-beta, interleukin 6 (IL6), C–X–C motif chemokine ligand 10 (CXCL10), tumor necrosis factor (TNF)-alpha, and interferon (IFN)-gamma by T-cells in the human CF islets [40,54], which are known to impair the secretory function of islets [55]. In fact, the isolation of islets from the inflammatory environment limited the effect of these inflammatory mediators, determining a normal in vitro secretion of insulin and glucagon [40].

Oxidative stress, a common feature of CF and diabetes [56,57], also seems to be involved in CFRD pathogenesis. Ntimbane et al. highlighted that defective CFTR expression or function can render beta-cells more susceptible to oxidative stress, thus contributing to endocrine cell dysfunction and to CFRD [58,59,60].

Incretin hormones glucose-dependent insulinotropic polypeptide (GIP) and glucagon-like peptide-1 (GLP-1) are also implicated in the development of CFRD. These hormones, released from the gut in response to nutrient ingestion, are responsible for 70% of insulin release after an oral or enteral glucose load [61]. Some studies reported that CFRD patients have lower GIP and GLP-1 levels than patients without diabetes and control groups [62], and this could contribute to the postprandial glycemic variability present in CFRD, associated with impaired first-phase insulin release [63].

Another factor that was described to have a role in CFRD onset is the accumulation of amyloid deposits [64]. Amyloid, a peptide normally produced by beta-cells [65], seems to exhibit toxicity toward these cells, leading them to death [66].

Insulin deficiency is the primary defect in CFRD, but the contribution of insulin resistance should also be considered. The contribution of insulin sensitivity to CFRD physiopathology is debated; in fact, conserved [67,68], improved [69], and reduced [70,71] insulin sensitivity have all been observed in CF. Several factors including chronic or recurrent respiratory infections, corticosteroid therapy, subclinical liver fibrosis, elevated serum levels of counter regulatory hormones, increased proinflammatory cytokines, and glycotoxicity could impact insulin sensitivity in patients with CF [3,40]. Overall, the data seem to suggest that, while insulin secretory defects are present in all people with CF, insulin resistance may occur only in subsets of individuals with CF or during distinct phases of the disease [67,72,73].

It is evident that the development of CFRD is influenced by several factors that act dynamically and simultaneously. Hypoglycemia in CF patients even without CFRD is a common finding, and, to date, there is still no unifying hypothesis for its etiology [74]. Spontaneous hypoglycemia can occur in adults or in children with CF, both in the fasting and the postprandial states, via different mechanisms. Some authors reported that fasting hypoglycemia in CF patients is due to malnutrition and possible underlying acute infection [33,34]; however, the association with malnutrition was not confirmed by other studies [75,76]. Indeed, the etiology of postprandial reactive hypoglycemia in CF was attributed to delayed, inadequate, or extended insulin responses in conjunction with delayed glucagon release [21,77]. Battezzati et al. hypothesized a dysregulation of basal insulin secretion related to the *CFTR* genotype as a possible cause of fasting hypoglycemia in CF [75]. In addition, as CFTR is also expressed in alpha-cells, mutations in the *CFTR* gene could also determine a dysregulated glucagon secretion in CF [78], thus contributing to hypoglycemia, although these data are controversial [40]. It has been hypothesized that reactive hypoglycemia could be a precursor to the development of CFRD [74]. Conversely, Radike et al. found that hypoglycemia did not predict a risk of development of CFRD or impaired glucose tolerance [79].

In addition to pancreatic fibrotic damage, determined largely by CFTR dysfunction [80], other factors may contribute to development of diabetes in CF. Known risk factors for CFRD include female sex, advancing age, lung function, liver disease, steroid treatment, family history of T2D, and genetic factors including both the *CFTR* gene and other modifier genes [81,82,83,84,85,86]. Genetic factors increase the prevalence of CFR [82,87], although it is not entirely clear whether they do so independently from other risk factors. This review aims to provide an overview on genetic factors associated with glucose abnormalities in CF.

## 2. Method of Search

A systematic literature search was performed using the following relevant keywords and search strings on PubMed, Science Direct, Scopus, and Google Scholar databases to identify studies and reports on glucose metabolism abnormalities in CF: “cystic fibrosis-related diabetes”, “glucose disorders”, “hypoglycemia”, “hyperglycemia”, “cystic fibrosis”, “insulin resistance”, “insulin secretion”, CFRD pathogenesis”, CFRD diagnosis”, “CFRD genetics”, and “CFRD and gene modifiers”.

Analyzed articles were selected by three authors independently, and in a second step these articles were supervised by three others. All authors reviewed and edited the manuscript.

## 3. Genetic Factors

### 3.1. CFRD and CFTR Genotypes

Functional studies have demonstrated that the *CFTR* genotype correlates with the severity of many CF manifestations. The onset of diabetes in CF is highly influenced by the specific *CFTR* genotype; in fact, a higher prevalence of diabetes was described in patients with more severe *CFTR* genotypes [88], unlike mutations causing mild CFTR dysfunction [30,89]. Several studies found that impaired glucose metabolism (prediabetes and diabetes) is a frequent comorbidity in patients with the F508del mutation both in the homozygous and in the heterozygous states [73,90]. Patients homozygous for the F508del mutations seem to have lower peak insulin concentrations at OGTT, lower first-phase insulin response (FPIR) during the intravenous glucose tolerance test (IVGTT), and lower insulin sensitivity compared to heterozygotes [73].

It was reported that a percentage of patients having the same *CFTR* genotype develop diabetes, and that, even among CFRD individuals with identical severe *CFTR* genotypes, there may be wide variation in the onset of diabetes [85].

*CFTR* genotype may predispose one to diabetes through exocrine pancreatic dysfunction or through a more direct role on islets. Pancreatic insufficiency correlates with the *CFTR* genotype, even if not all patients with pancreatic exocrine disease develop diabetes [85]. The defect of CFTR could itself be the cause of impaired glucose metabolism. In fact, as reported above (see Section 1.3), CFTR is also involved in pancreas development and plays an intrinsic role in beta- and alpha-cells [49,50], influencing both insulin and glucagon secretion. Furthermore, CFTR dysfunction makes beta-cells more susceptible to oxidative stress with subsequent apoptosis of this cell line and development of diabetes [40].

As severe *CFTR* genotype increases the risk of diabetes independently of pancreatic exocrine dysfunction [30], a screening for prediabetes is mandatory in patients with two severe *CFTR* mutations, particularly F508del, and insulin supplementation seems a rational therapy to consider [90,91].

The recent development of novel drugs capable of correcting the basic CFTR defect promises to direct the CF therapeutic chances toward personalized medicine designed to overcome specific mutational abnormalities [92]. In particular, more severe mutations (classes I and II), which reduce protein quantity, could be treated by corrector drugs, while, for other mutations (classes III, IV, and V) retaining a residual protein function, potentiators could be useful [93]. Lumacaftor (VX-809) is an investigational CFTR corrector that increases the number of channels present in the plasma membrane and improves CFTR-mediated chloride transport in vitro [94]. It was shown that VX-809 also corrects glucose-induced electrical abnormalities and insulin secretion in beta-cells with the F508del mutation, opening the possibility for clinical use of this drug in CFRD [50]. With regard to CFTR potentiators, in CF patients with at least one CFTR conductance or gating mutation, treatment with ivacaftor significantly improved their insulin response, particularly during the early phase of insulin secretion, suggesting that these drugs could improve glucose tolerance in CF patients [50,51]. On the basis of these results, it was hypothesized that early use of a CFTR corrector in very young children, who still have the presence of considerable beta-cell mass, or even before birth, could delay or prevent the development of diabetes [51,52,95]. However, longer-term clinical studies involving larger numbers of patients are needed to confirm the proposed benefit of these drugs in treating or preventing CFRD [95]. These data suggest that an effort is needed to expand the use of CFTR modulators even for genotypes for whom it has not yet been approved, through an ex vivo evaluation of the pharmacological response of patients is currently excluded from treatment [96,97].

### 3.2. CFRD and Modifier Genes

Genetics studies on twins and siblings with CF have demonstrated that the risk of diabetes correlates with the degree of gene-sharing among related CF patients, indicating that genetic factors different from the *CFTR* genotype contribute to the development of CFRD [85].

CFRD and T2D show similar clinical and pathophysiological features, and both have a strong genetic bases [98]. In fact, a family study highlighted that a family history of T2D increases the risk of diabetes in individuals with CF and that a variant in the transcription factor 7-like 2 (*TCF7L2*) gene, known to be associated with T2D prevalence in the general population, is also involved in CFRD. In particular, the *rs7903146 TCF7L2* variant increases the risk of diabetes threefold and decreases the age of onset by 7 years [85]. The *TCF7L2* gene encodes for a transcription factor belonging to the Wnt signaling pathway expressed in fat, liver, and pancreatic islets, where its function is to regulate insulin production and processing [99,100,101,102]. In T2D patients and in healthy subjects with the *rs7903146 TCF7L2* variant, impaired insulin secretion and defects in glucagon suppression were reported, as well as abnormal insulin processing [99,103,104].

This genetic variant was also evidenced in T1D individuals with a single autoantibody more frequently with respect to individuals with the presence of multiple autoantibodies, suggesting that the *TCF7L2* locus may also play a role in the development of diabetes in subjects with a limited autoimmunity [105]. In addition, because individuals with single autoantibodies have a lower risk of developing T1D with respect to those with multiple autoantibodies, it is possible that these subjects, in the absence of this *TCF7L2* variant, may not develop overt diabetes [105].

Other studies suggested a possible regulatory role of the *TCF7L2* variant in expression of the *ACSL5* gene, encoding acyl-CoA synthetase long-chain family member 5, essential for fatty acid metabolism, which could be involved in insulin sensitivity [106,107].

Other authors evidenced that the *TCF7L2* variant is located in a binding site of the *FOXA2* gene, encoding a transcriptional factor that is involved in pancreatic and hepatic development [108], as well as in islet-selective open chromatin regions that regulate gene expression [109], allowing them to hypothesize a possible role of this variant in the early onset of diabetes.

In 2013, Blackman et al., using genome-wide association and candidate gene-based approaches, confirmed association with the *TCF7L2* gene and identified four new loci associated with CFRD: *CDKAL1*, *CDKN2A/B*, *IGF2BP2*, also associated with T2D [110], and *SLC26A9*, specific to CFRD [110]. These five loci were estimated to account for 8.3% of the phenotypic variance in CFRD onset and had a combined population-attributable risk of 68% [85,110].

Cyclin-dependent kinase 5 (CDK5) regulatory subunit-associated protein 1-like (CDKAL1) is required for normal mitochondrial morphology and function in pancreatic islets and other tissues such as adipose tissue, which can contribute to the pathogenesis of type 2 diabetes. Variants in *CDKAL1* are reported to impair proinsulin translation and to stimulate the endoplasmic reticulum stress response [111], which promotes apoptosis [112].

Cyclin-dependent kinase inhibitors 2A and 2B (CDKN2A/B) regulate two cell-cycle regulation pathways, the p53 pathway and the retinoblastoma 1 (RB1) pathway [113]. CDKN2A/B are tumor suppressors with reported roles in both cellular senescence and insulin secretion; in fact, they are related to the reduced proliferation and regeneration of beta-cells, and they reduce the secretory function of insulin [114].

Insulin-like growth factor 2 messenger RNA (mRNA)-binding protein 2 (IGF2BP2), highly expressed in pancreatic islets, belongs to the family of insulin-like growth factor 2 (IGF2) mRNA-binding proteins, which play roles in normal embryonic growth and development [115]. Moreover, *IGF2BP2* was found to be associated with decreased insulin secretion [116].

Solute carrier family 26 member 9 (SLC26A9) is a member of the solute-linked carrier 26 (SLC26) anion transporter family [117]. Two variants in the promoter and in the first intron of the *SLC26A9* gene were reported to increase risk and to influence the age of onset of CFRD [110]. SLC26A9 represents an alternative chloride channel involved in ion and fluid secretion in several epithelial tissues, including airways and pancreas. Therefore, considering its function as a chloride channel, it is possible that it also regulates membrane potential and insulin secretion in beta-cells [117]. Another variant identified in association with CFRD is located in intron 5 of the *SLC26A9* gene. This variant was described in association with pancreatic insufficiency and early exocrine pancreatic disease in CF patients [118]. Expression studies showed that SLC26A9 and CFTR interact reciprocally under both physiological and pathophysiological conditions [117,119]. Other studies suggested that SLC26A9 may be able to compensate for the CFTR dysfunction in CF patients [117]. The evidence of *SLC26A9*′s role as a gene modifier and of its coexpression with CFTR in several organs involved in CF suggests that therapeutic strategies increasing the level and/or function of SLC26A9 could represent an alternative method to bypass the defect of ion transport [117], which would be useful not only as therapy in CF but also for the treatment and prevention of CFRD [117].

CFRD variants at *SLC26A9* are also significantly associated with the expression of a neighboring gene, *PM20D1*. This gene, located 63 kb downstream of *SLC26A9*, encodes peptidase M20 domain-containing 1, a secreted enzyme involved in the regulation of energy expenditure in adipocytes, where it determines a reduction in fat mass and in glucose [120]. On the basis of their functions, *PM20D1* and *SLC26A9* are considered plausible candidate genes for CFRD modifiers.

In a very recent study on 5740 individuals with CF, Aksit et al. identified a novel genetic modifier, *PTMA*, located on chromosome 2, and they replicated the previous association of variants at the *TCF7L2* and *SLC26A9* loci [121]. *PTMA* encodes for prothymosin-α (ProT), which has a role in inflammation, oxidative stress, cell proliferation, and apoptosis. Furthermore, PTMA may also affect insulin sensitivity by acting as a ligand for Toll-like receptor 4 [122,123].

The variants at the *PTMA* locus are also situated within and near the *PDE6D* and *COPS7B* genes. *PDE6D* encodes for the delta subunit of rod-specific photoreceptor phosphodiesterase, a key enzyme involved in the phototransduction cascade. COPS7B is a component of the COP9 signalosome complex involved in various cellular and developmental processes, and it is an essential regulator of the ubiquitin pathway [121]. In addition to the abovementioned genes, other additional loci emerged in the same study that influence both CFRD and T2D (Table 3). All these variants seem to affect beta-cell function and insulin secretion [121].

A relationship between CFRD and genes associated with inflammation such as tumor necrosis factor [86], heat-shock protein [136], and calpain 10 [137] was described.

In a very recent study, Pineau et al. analyzed the transcriptome in blood samples from CF patients to identify genes and pathways that modulate the comorbidities associated with CF. Results of this study highlighted genes that belong to relevant biological pathways, such as cell adhesion, leukocyte trafficking, and production of reactive oxygen species, as being central in lung function decline and cystic fibrosis-related diabetes [20].

It is necessary to take into account that not all hyperglycemia cases in CF can be related to this pathology, and other etiologies need to be considered. Recently, four atypical cases of young CF patients were reported with hyperglycemia associated with mutations in the hepatocyte nuclear factor 1-alpha (*HNF1A)* (two siblings) and Glucokinase (*GCK)* (two single cases) genes responsible for the two more frequent forms of monogenic diabetes [138,139,140,141,142]. Monogenic diabetes, which accounts for 1–6% of pediatric diabetes cases, comprises a group of heterogeneous genetic disorders characterized by early onset of diabetes, absence of autoimmunity, and beta-cell dysfunction [143]. Recognition of these forms of diabetes is crucial in selecting the most appropriate treatment and follow-up for patients [144]. In particular, HNF1A maturity-onset diabetes of the young (MODY) is very amenable to therapy with sulfonylureas, whereas, for GCK MODY, no pharmacological treatment is recommended, and diet and regular physical activity are necessary to maintain good glycemic control [140]. It is evident that, although unlikely, the possibility of monogenic forms of diabetes in CF patients should be considered in the diagnostic process, particularly in children aged below 10 years. In this case, the coexistence of two genetic defects, which can determine beta-cell dysfunction in different manners, necessitate a careful monitoring of the glycemic profile, because it may be very crucial for the treatment plan of the patient.

## 4. Concluding Remarks

In conclusion, despite recent advances, knowledge on CFRD pathogenesis, pathophysiology, and treatment remains incompletely defined. Insulin deficiency is recognized as a primary defect; however, additional factors, including genetic factors, seem to play important roles.

The studies published so far underline the importance of understanding the genetic architecture of CFRD to better define the pathogenesis of the disease, to clarify the differences in clinical and metabolic characteristics, to predict its evolution, and to identify individualized treatment decisions among affected patients. In fact, the recent discoveries of novel therapeutic approaches based on CFTR corrector/modulator drugs and the hypotheses related to other possible strategies that increase the level and/or function of other proteins involved in CFRD pathogenesis, to be used early in young patients, have led to a new scenario in the treatment and prevention of diabetes in CF. However, there are still many unresolved questions regarding the real efficacy of these drugs; therefore, much research will be needed in this field.

## Figures and Tables

**Figure 1 diagnostics-11-00572-f001:**
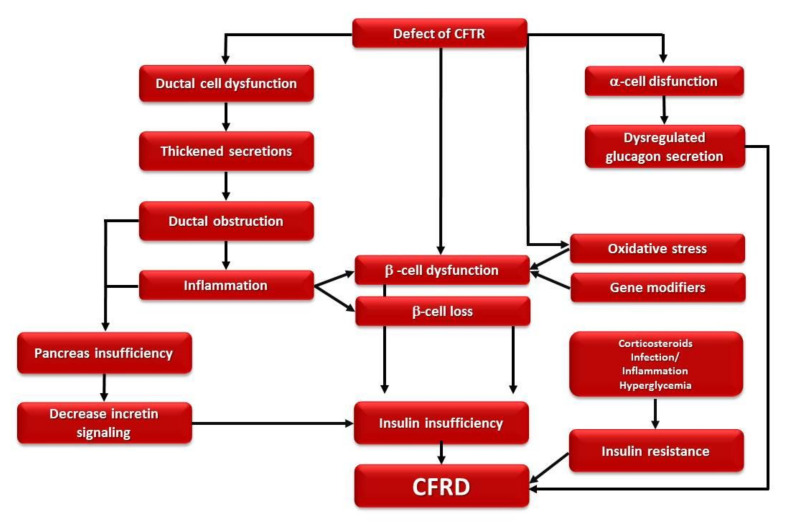
Diagram showing the CFRD pathogenesis.

**Table 1 diagnostics-11-00572-t001:** Characteristics of the different forms of diabetes: type 1 diabetes, type 2 diabetes, and cystic fibrosis-related diabetes (CFRD). Adapted from Moran et al. [28].

	Type 1 Diabetes	Type 2 Diabetes	CFRD
Prevalence in CF	0.2%	11%	35%
Onset	Acute	Insidious	Insidious
Age of onset	Children, adolescents	Adults	18–24 years
Weight	Normal	Obesity/overweight	Normal/underweight
Autoimmune pathogenesis	Yes	No	No
Insulin deficiency	Almost total	Partial	Severe but not total
Insulin sensitivity	Normal/slightly decreased	Severely decreased	Variably decreased (>acute phases)
Ketones	Yes	Rare	Rare
Therapy	Insulin	Diet Oral hypoglycemic agents Insulin	Insulin
Microvascular complications	Yes	Yes	Yes
Macrovascular complications	Yes	Yes	No
Metabolic syndrome	No	Yes	No
Cause of death	Cardiovascular	Cardiovascular	Pulmonary

**Table 2 diagnostics-11-00572-t002:** Interpretation of oral glucose tolerance test (OGTT) in cystic fibrosis.

	Fasting		2 h Glucose		Notes
	mg/dL	mmol/L	mg/dL	mmol/L	
Normal glucose tolerance (NGT)	<126	<7	<140	<7.8	All glucose levels <200 mg/dL (11.1 mmol/L)
Indeterminate glycemia (INDET)	<126	<7	<140	<7.8	Mid-OGTT glucose ≥200 mg/dL (11.1 mmol/L)
Impaired glucose tolerance (IGT)	<126	<7	140–199	7.8–11	
CFRD with fasting hyperglycemia	≥126	≥7	≥200	≥11.1	
CFRD without fasting hyperglycemia	<126	<7	≥200	≥11.1	

**Table 3 diagnostics-11-00572-t003:** List of additional genetic modifiers of CFRD overlapping with type 2 diabetes (T2D) [121].

Symbols	Gene	Locations	Protein (https://www.genecards.org/ (Accessed on 3 November 2020)
*ADCY5*	Adenylate cyclase 5	Chr 3	Component of the membrane-bound adenylyl cyclase enzymes. Regulates the increase in free cytosolic Ca^2+^ in response to increased blood glucose levels, contributing to insulin secretion [124].
*ANK1*	Ankyrin 1	Chr 8	Ankyrins play key roles in cell motility, activation, proliferation, contact, and maintenance of specialized membrane domains [125].
*BCAR1*	BCAR1 scaffold Protein, Cas family member	Chr 16	Component of the Crk-associated substrate (Cas) family of scaffold proteins, involved in several cellular pathways, including cell motility, apoptosis, and cell-cycle control [126].
*CEBPB*	CCAAT enhancer-binding protein beta	Chr 20	Transcription factor that regulate genes involved in immune and inflammatory responses [127].
*DGKB*	Diacylglycerol kinase beta	Chr 7	Diacylglycerol kinases metabolize 1,2,diacylglycerol to produce phosphatidic acid, important for cellular processes [128].
*ETS1*	ETS proto-oncogene 1, transcription factor	Chr 11	Component of the ETS family of transcription factors that controls the expression of cytokine and chemokine genes, the differentiation, survival, and proliferation of lymphoid cells, and angiogenesis [129].
*GLIS3*	GLIS family zinc finger 3	Chr 9	Component of the GLI-similar zinc finger protein family involved in the development of pancreatic beta-cells, thyroid, eye, liver, and kidney [130].
*LTK*	Leukocyte receptor tyrosine kinase	Chr 15	Component of the ros/insulin receptor family of tyrosine kinases, very important for cell growth and differentiation [131].
*MAEA*	Macrophage erythroblast attacher, E3 ubiquitin ligase	Chr 4	Protein that mediates the attachment of erythroblasts to macrophages. It is required for normal cell proliferation and may contribute to nuclear architecture and cell division events [132].
*SHQ1*	SHQ1, H/ACA ribonucleoprotein assembly factor	Chr 3	Protein that assists in the assembly of H/ACA-box ribonucleoproteins involved in the processing of ribosomal RNAs, modification of spliceosomal small nuclear RNAs, and stabilization of telomerase [133].
*SLC2A2*	Solute carrier family 2 member 2	Chr 3	Integral plasma membrane glycoprotein important for the bidirectional glucose transfer across the plasma membrane of hepatocytes, beta-cells, intestine, and kidney epithelium [134].
*SLC30A8*	Solute carrier family 30 member 8	Chr 8	Zinc efflux transporter expressed at a high level only in the pancreas, involved in insulin maturation and/or in insulin secretion [135].

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
