# Peer review of "Cystic Fibrosis-Related Diabetes (CFRD): Overview of Associated Genetic Factors"

_diagnostics, 2021, doi:10.3390/diagnostics11030572_

Round 1

Reviewer 1 Report

Dear Authors,

thank you for your responses to my answers. The answers are clear and corrected.

I have no further comments to do about the revised Manuscript.

Best regards

Reviewer 2 Report

The reviewer thanks the authors for taking the time to consider each concern. The reviewer has no additional comments or concerns.

This manuscript is a resubmission of an earlier submission. The following is a list of the peer review reports and author responses from that submission.

Round 1

Reviewer 1 Report

Dear Authors,

I would congratulate you for the excellent work and the huge job you have done.I think is a very nice review covering a field of great interest.

I have some minor comments:

  • About the methodology of the review: pleas state if the intention was for a systematic review or for a narrative review.
  • Who reviewed the articles after the searches? Usually the correct way is that two reviewers  analyze the articles extracted indipendentely, and a referee eventually revise what is not in accordance between the reviewers.
  • Once more, please provide a flow chart and checklist,of the included/excluded articles, according to PRISMA indications

Best regards,

Reviewer 2 Report

Major Comments. This review lacks a central rationale, e.g., how is this overview different than those already published? A focus on genetic modifiers would provide a central focus and more unique review. However, in its current form this section is only a minor part of the review and only slightly more detailed than a table.

Line 7; Use an institutional email address

Line 16; 'on membrane' to 'on the membrane'

Line 17; 'most common' to 'a common'

Line 18; 'a distinct form' is followed by a general description of diabetes

Line 34; change 'Cystic fibrosis transmembrane regulator' to 'cystic fibrosis transmembrane conductance regulator'.

Line 37-40; may be an over simplification

Line 57; Modulators are drugs. I think the authors mean modifiers

Table 1; The prevalence for each may need clarification for example CFRD (35% of CF). For diabetes, are these world values or specific to a country.

Line 88; it is unclear what is being said

Line 93; OGTT is not defined until line 113

Line 95; it is not clear what defines a severe genotype. If severity is a measure of residual CFTR function, then severe genotypes are also pancreatic insufficient

Line 96; very young is a generalization. What ages why is this considered very young?

Line 98; not sure what the authors are trying to convey. what should be suspected (CF?). This statement may be dependent on the CF screening methods of the patient's country.

Line 125; define ISPAD

Line 171+; This is speculative. Ref 48, says that Beta cells accumulate Cl- above equilibrium potential through NKCC1 and give a Cl dose-response. It does not suggest that intracellular Cl is altered in CF.

Line 171+; Based on Ref49 experiments, the simple explanation is that the transference number for Cl decreases in CF. Alterations in intracellular Cl- were not tested in CF, only the acute effect of a CFTR inhibitor. Results could have been due to a combination of resting membrane potential and/or cellular input resistance.

Line 174; correctors/potentiators restore CFTR function rather quickly (correction in sweat Cl seen as early as 48hrs). Therefore, it'd be useful to know the timing of the findings referenced around line 174.

Line 177; how is CFTR activated by glucose?

Line 178 and other areas; reformat KATP with ATP subscripted

Line 178; none of the provided citations supports that Cl flux potentiates the K channel, rather it is likely the CFTR channels contribution to RMP and input resistance.

Line 180 and other areas; reformat Ca2+ appropriately

Paragraph 273-276 this statement could be country-dependent; please verify

Line 285-286; if these drugs are being given to CF patients, then wouldn't they be treating CFRD in the genotypes that they are approved for?

Section 3.2 this section doesn’t provide much information beyond being a narrative of a table.